# Magnetized Particle Motion around Black Holes in Conformal Gravity: Can Magnetic Interaction Mimic Spin of Black Holes?

**Kamoliddin Haydarov [1], Ahmadjon Abdujabbarov [1,2,3,4,5]** and **Javlon Rayimbaev [3,5,6]** and **Bobomurat Ahmedov [3,4,5,*]**

[1] Department of Physics, Tashkent University of Information Technologies named after Muhammad al Khwarizmi, Amir Temur 108, Tashkent 100014, Uzbekistan; haydarovk378@gmail.com (K.H.); ahmadjon@astrin.uz (A.A.)

[2] Shanghai Astronomical Observatory, 80 Nandan Road, Shanghai 200030, China

[3] Ulugh Beg Astronomical Institute, Astronomicheskaya 33, Tashkent 100052, Uzbekistan; javlon@astrin.uz

[4] Department of Physics and Chemistry, Tashkent Institute of Irrigation and Agricultural Mechanization Engineers, Kori Niyoziy 39, Tashkent 100000, Uzbekistan

[5] Physics Faculty, National University of Uzbekistan named after Mirzo Ulugbek, University 4, Tashkent 100174, Uzbekistan

[6] Laboratory of Theoretical Nuclear Physics, Institute of Nuclear Physics, Ulugbek 1, Tashkent 100214, Uzbekistan

[*] Correspondence: ahmedov@astrin.uz

**Abstract:** Magnetized particle motion around black holes in conformal gravity immersed in asymptotically uniform magnetic field has been studied. We have also analyzed the behavior of magnetic fields near the horizon of the black hole in conformal gravity and shown that with the increase of conformal parameters $L$ and $N$ the value of angular component of magnetic field at the stellar surface decreases. The maximum value of the effective potential corresponding to circular motion of the magnetized particle increases with the increase of conformal parameters. It is shown that in all cases of neutral, charged and magnetized particle collisions in the black hole environment the center-of-mass energy decreases with the increase of conformal parameters $L$ and $N$. In the case of the magnetized and negatively charged particle collisions, the innermost collision point with the maximum center-of-mass energy comes closer to the central object due to the effects of the parameters of the conformal gravity. We have applied the results to the real astrophysical scenario when a pulsar treated as a magnetized particle is orbiting the super massive black hole (SMBH) Sgr A$^*$ in the center of our galaxy in order to obtain the estimation of magnetized compact object's orbital parameter. The possible detection of pulsar in Sgr A$^*$ close environment can provide constraints on black hole parameters. Here we have shown that there is degeneracy between spin of SMBH and ambient magnetic field and consequently the interaction of magnetic field $\sim 10^2$ Gauss with magnetic moment of magnetized neutron star can in principle mimic spin of Kerr black holes up to 0.6.

**Keywords:** black hole; magnetized particle motion; magnetic field; conformal gravity; constraints on black hole parameters

## 1. Introduction

One of the fundamental problems of general theory of relativity is the presence of singularity in almost all known exact analytical solutions of the gravitational field equations. For the black hole solutions, the central physical singularity with the infinite curvature is unavoidable. However

standard understanding of the physics cannot accept the physical processes at the physical singularity, and it breaks out. There are several attempts to avoid the singularity: coupling with nonlinear electrodynamics [1–7], conformal transformations [8–13], quantum gravity corrections [14–16], etc.

One of the possible ways of excluding the physical singularity in the black hole solutions is using the conformal gravity where metric tensor is transformed as

$$g_{\mu\nu} \to g_{\mu\nu}^* = \Omega^2 g_{\mu\nu}\,, \tag{1}$$

where $\Omega = \Omega(x)$ is a conformal factor of transformation.

Using the modification of Einstein's gravity by the auxiliary scalar field $\phi$ (dilaton) one may obtain the following Lagrangian for gravity

$$\mathcal{L}_1 = \phi^2 R + 6\,g^{\mu\nu}(\partial_\mu\phi)(\partial_\nu\phi)\,. \tag{2}$$

Other efficient way of introducing conformal gravity without introducing dilaton can be performed via following Lagrangian

$$\mathcal{L}_2 = a\,C^{\mu\nu\rho\sigma}C_{\mu\nu\rho\sigma} + b\,{}_*R^{\mu\nu\rho\sigma}R_{\mu\nu\rho\sigma}\,. \tag{3}$$

where $C^{\mu\nu\rho\sigma}$ is the Weyl tensor, $R^{\mu\nu\rho\sigma}$ is the Riemann tensor, ${}_*R^{\mu\nu\rho\sigma}$ is the dual of the Riemann tensor, $a$ and $b$ are constants.

In Einstein's theory of gravity, the singularity can be resolved by suitable conformal transformation if a space-time metric $g_{\mu\nu}$ is singular in a gauge. Singularity-free black hole solutions in conformal gravity have been proposed in Refs. [12,13]. It was shown that these spacetimes are geodetically complete because no massless or massive particles can reach the center of the black hole in a finite amount of time or for a finite value of the affine parameter [12,13]. Within this theory the curvature invariants do not diverge at the center $r = 0$.

The electromagnetic fields of slowly rotating neutron stars in conformal gravity have been studied in [17]. The authors of Ref. [18] have tested the conformal gravity with the SMBH observation. The energy conditions for conformal gravity are studied in [19] while scalar perturbations of non-singular non-rotating black holes in conformal gravity have been studied in [20].

Here we plan to study the electromagnetic field structure and magnetized particle motion around black holes in conformal gravity immersed in external magnetic field.

The black hole immersed in magnetic field has been studied in the pioneering work of Wald [21]. After that, numerous studies have been devoted to the electromagnetic field around various black holes and particle motion in external magnetic field (see e.g.,) [22–41].

In particular, the motion of particles with non-zero spin and magnetic dipole momentum around black holes in external magnetic field has been studied in [42]. It was also shown that magnetized particles can move along stable non-geodesic, spatially circular equatorial orbits with the radius smaller than innermost stable circular orbits (ISCO) (in the case of non-rotating black holes) and close to photon orbit. The magnetized particle motion around rotating black holes has been studied in [43].

Magnetized particle motion around non-Schwarzschild black holes immersed in an external uniform magnetic field has been recently studied in [44]. Acceleration of magnetized particles around rotating black holes in quintessence and high energy collision of magnetized particles around a Hořava–Lifshitz black hole have been studied in [29,30]. Magnetized particles acceleration around a Schwarzschild black hole in a magnetic field has been analyzed in [32]. Magnetized particle motion around a Braneworld black hole has been considered in [45,46].

The energetic processes around the black hole through the different mechanisms of black hole energy extraction are the subject of interest of modern relativistic astrophysics. The Penrose process [47], Blandford-Znajeck mechanism [48], Magnetic Penrose process [49–52] and particle acceleration mechanism (BSW) [53] are several of them which can be used for modelling the high energetic

processes of active galactic nuclei and other X-ray sources. The energetic processes around black holes in different scenarios in alternative and modified theories of gravity have been extensively studied e.g., in [27,29,31–34,36,54–56].

This work is devoted to study the magnetized particle motion and acceleration processes around non-rotating compact object in conformal gravity in the presence of magnetic field and organized as follows: In Section 2 we study the electromagnetic field configuration around black holes immersed in external asymptotically uniform magnetic field in the framework of the conformal gravity. Section 3 is devoted to study of the magnetized particles motion around black holes in conformal gravity. Magnetized particles acceleration in non-singular black holes immersed in an external magnetic field is studied in Section 4. Section 5 is devoted to the possible astrophysical applications to the degeneracy between the spin of SMBH Sgr A* and environment magnetic field. We summarize the obtained results in Section 6.

Throughout this work we use signature $(-,+,+,+)$ for the space-time and geometrized unit system $G = c = 1$ (However, for an astrophysical application we have written the speed of light and Newtonian constant explicitly in our expressions). Latin indices run from 1 to 3 and Greek ones from 0 to 3.

## 2. Electromagnetic Fields around Black Holes Immersed in Asymptotically Uniform Magnetic Field

General relativity according to no-hair theorem predicts that asymptotically uniform external magnetic field of a black hole is defined by mass $M$, electric charge $Q$ and specific angular momentum $a = J/M$, where is $J$ angular momentum.

If a black hole is charged with the electric charge $Q$ it creates the radial electric field $E^{\hat{r}} = Q/r^2$ and black hole's rotation generates additional dipolar magnetic field

$$B^{\hat{r}} = \frac{2\mathcal{M}}{r^3} \cos\theta \, , \qquad B^{\hat{\theta}} = \frac{\mathcal{M}}{r^3} \sin\theta \, , \tag{4}$$

where $\mathcal{M} = Qa$ is the magnetic moment of the black hole. However, the case of charged holes does not have realistic astrophysical interest because electrostatic forces which are much stronger than gravitational ones promptly neutralize the black hole's electric charge $Q$, attracting electric charges of opposite sign from the surrounding environment. For this reason, intrinsic magnetic moment of black holes $\mathcal{M}$ is equal to zero.

The authors of Ref. [57] have first shown that the magnetic moment of magnetized star vanishes during the gravitational collapse. Due to conservation of magnetic flux during the collapse $B = B_0(R/R_0)^2$, where $R_0$ and $B_0$ are initial values of radius and magnetic field of the star before the gravitational collapse. Then, in the non-relativistic Newtonian limit magnetic moment of the star $\mathcal{M} \sim BR^3$ decays as $\mathcal{M} = \mathcal{M}_0(R/R_0)^3$ and consequently $\mathcal{M} \to 0$ when $R \to 0$.

In general relativity during gravitational collapse the stellar magnetic moment decays as [57]

$$\mathcal{M}(t) = \mathcal{M}_0 \frac{4M^2}{3R_0 t} \, ,$$

and in the quasi-stationary regime the exterior magnetic field should decay with $t^{-1}$. The more correct faster decay rate as $t^{-(2l+2)}$ (here $l$ is multipolarity of electromagnetic radiation) taking into account electromagnetic radiation outside a black hole has been obtained in [58].

Even though the black hole does not have own magnetic field one may study the electromagnetic field around singularity-free non-rotating black holes immersed to external asymptotically uniform

magnetic field. In Schwarzschild coordinates $(t, r, \theta, \phi)$ the space-time metric of the spherically symmetric static black hole in conformal gravity can be described as [12,13]

$$ds^2 = S(r)\left[-f dt^2 + \frac{dr^2}{f} + r^2(d\theta^2 + \sin^2 d\phi^2)\right] , \tag{5}$$

where $f = 1 - 2M/r$ is the lapse function and the scaling factor $S(r)$ has the following form

$$S(r) = S = \left(1 + \frac{L^2}{r^2}\right)^{2N} , \tag{6}$$

with $N$ being a quantity describing conformal gravity assumed to be an integer, $L$ is a new conformal parameter of the black hole coming from the theory.

The vector potential of the electromagnetic field in the vicinity of a black hole can be found using Wald approach [21]. In the Lorentz gauge ($\nabla_\alpha A^\alpha = 0$) the Maxwell equations for the electromagnetic field potential $A^\alpha$ can be written as

$$\nabla_\mu \nabla^\mu A^\nu = 0 . \tag{7}$$

On the other hand, the Killing vector $\xi^\mu$ satisfies the following equation

$$\nabla_\mu \nabla^\mu \xi^\nu + R^\nu{}_\mu \xi^\mu = 0 , \tag{8}$$

for the given space-time. As one can see Equations (7) and (8) have the same mathematical form in the case of Ricci flat spacetimes ($R^\nu{}_\mu = 0$), which allows us to write the solution for the electromagnetic field potential as a linear combination of Killing vectors. From the existence of timelike $\xi^\alpha_{(t)} = (-1, 0, 0, 0)$ and spacelike $\xi^\alpha_{(\phi)} = (0, 0, 0, 1)$ Killing vectors for the space-time metric (5) one can write the solution of Equation (7) in the following form

$$A^\alpha = C_1 \xi^\alpha_{(t)} + C_2 \xi^\alpha_{(\phi)} , \tag{9}$$

where $C_1$ and $C_2$ are constants of integration. From the asymptotic properties of the space-time and the electromagnetic field one can easily find these constants to have the values $C_1 = 0$ and $C_2 = B/2$ (see, e.g., [33,36,59]), where $B$ is the asymptotic value of the uniform magnetic field.

Finally, the covariant components of the four-vector potential of the electromagnetic field read

$$A_0 = A_1 = A_2 = 0 , \quad A_3 = \frac{1}{2} B S r^2 \sin^2 \theta , \tag{10}$$

and the non-zero components of electromagnetic field tensor $F_{\mu\nu}$ are

$$F_{r\phi} = \frac{r^2 + (2N+1)L^2}{r^2 + L^2} B S r \sin^2 \theta , \tag{11}$$

$$F_{\theta\phi} = B S r^2 \sin\theta \cos\theta . \tag{12}$$

The non-zero orthonormal components of the magnetic field in the frame of static observer take the following form

$$B^{\hat{r}} = B \cos\theta , \tag{13}$$

$$B^{\hat{\theta}} = \left(1 - \frac{2NL^2}{L^2 + r^2}\right) \sqrt{f} B \sin\theta . \tag{14}$$

One can see from Equations (13) and (14) that the radial component of the magnetic field does not depend on conformal parameter while the $\theta$-component does.

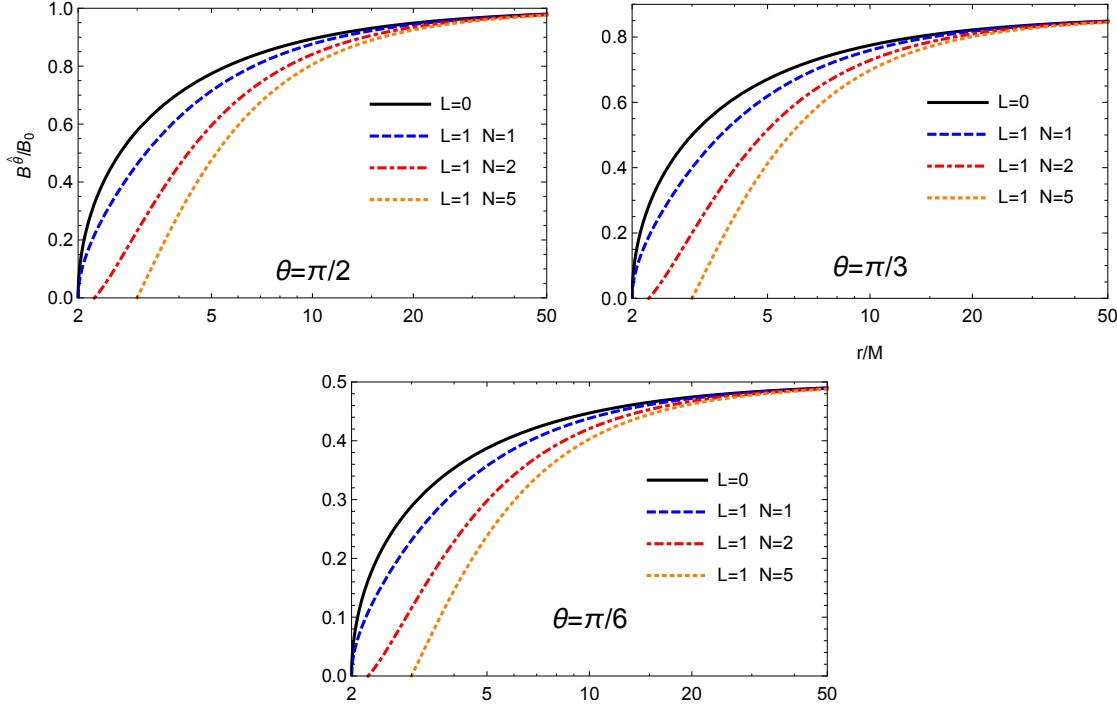

**Figure 1.** Radial dependence of normalized angular component of magnetic field for different angles with different values of conformal parameters $L$ and $N$. The **top left panel** is for $\theta = \pi/2$, the **top right one** is for $\pi/3$ and the **bottom one** is for $\theta = \pi/6$.

In Figure 1 we have shown the radial dependence of the $\theta$-components of external uniform magnetic field around singularity-free black holes. One can see from the figures that the value of the $\theta$-component decreases with the increase of the value of the parameter $N$ for the fixed value of $L$ while radial one goes up exponentially in the black hole proximate environment. As expected at far distances from a black hole, magnetic field components tend to the Newtonian limit. The structure of the magnetic field in the vicinity of a black hole is shown in Figure 2 for the various values of conformal gravity parameters $L$ and $N$.

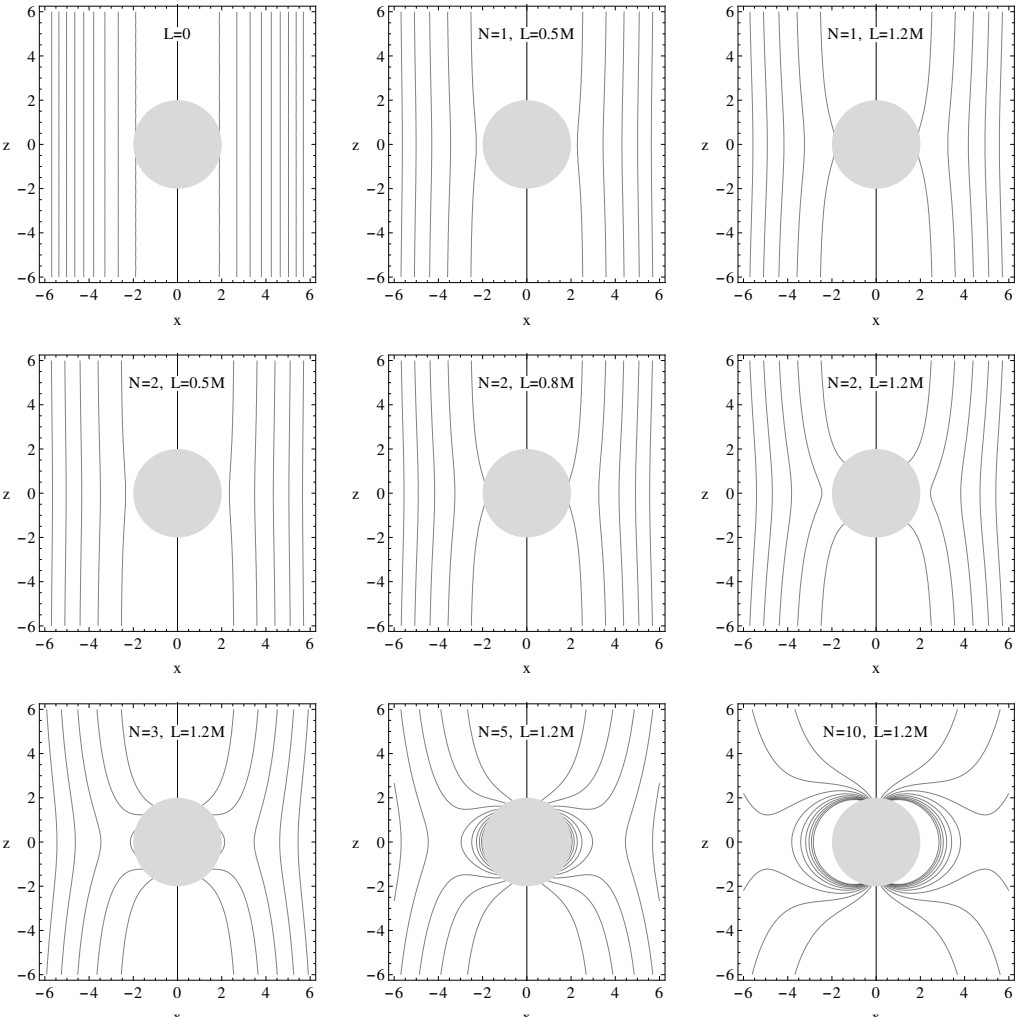

**Figure 2.** Profiles of the magnetic field lines for the different values of the conformal parameters $L$ and $N$ on $z - x$ plane (where new coordinates are defined as $z = r \cos \theta$ and $x = r \sin \theta$).

## 3. Magnetized Particle Motion around Black Holes in Conformal Gravity

In this section, we will study motion of particles with non-zero magnetic moment around black holes in conformal gravity immersed in an external asymptotically uniform magnetic field. We use Hamilton-Jacobi equation to describe motion of the magnetized particles. The Hamilton-Jacobi equation of motion of magnetized particles can be written in the following form [42]

$$g^{\mu\nu} \frac{\partial S}{\partial x^\mu} \frac{\partial S}{\partial x^\nu} = - \left( m - \frac{1}{2} D^{\mu\nu} F_{\mu\nu} \right)^2 , \tag{15}$$

where $m$ is mass of a charged particle, $D^{\mu\nu} F_{\mu\nu}$ characterizes the interaction of external magnetic field with magnetized particle. Tensor $D^{\alpha\beta}$ can be expressed in the following form [42]

$$D^{\alpha\beta} = \eta^{\alpha\beta\sigma\nu} u_\sigma \mu_\nu , \qquad D^{\alpha\beta} u_\beta = 0 , \tag{16}$$

where $\mu^\nu$ is the magnetic dipole moment, $\eta^{\alpha\beta\sigma\nu}$ is Levi Civita tensor and $u^\nu$ is four-velocity of the particle. The electromagnetic field tensor $F_{\alpha\beta}$ can be decomposed through electric $E_\alpha$ and magnetic $B^\alpha$ fields in the following form

$$F_{\alpha\beta} = u_{[\alpha} E_{\beta]} - \eta_{\alpha\beta\sigma\gamma} u^\sigma B^\gamma . \tag{17}$$

Now using Equations (16) and (17) one can easily calculate the interaction term $D * F$ in the following form

$$D^{\alpha\beta}F_{\alpha\beta} = 2\mu^{\alpha}B_{\alpha} = 2\mu B_0 \mathcal{L}[\lambda_{\hat{\alpha}}] \,, \tag{18}$$

where $\mu$ is the module of magnetic moment of the particle and $L[\lambda_{\hat{\alpha}}]$ is a function of radial coordinate and defines the tetrad $\{\lambda_{\hat{\alpha}}\}$ attached to the co-moving fiducial observer (e.g., the orbital angular velocity of the particle).

Consider the orbital motion of magnetized particles around black holes in conformal gravity immersed in external uniform magnetic field in the approximation of neglecting higher order terms of magnetic momentum - electromagnetic field interaction $((D \cdot F)^2 \to 0$ limit). This corresponds to the case when interaction between external magnetic field and magnetized particle is weak due to test magnetic field or small value of the magnetic moment. At the equatorial plane $(\theta = \pi/2)$ the equations of motion can be expressed using conservative quantities as

$$\dot{t} = \frac{\mathcal{E}}{f \, S} \,, \tag{19}$$

$$\dot{r}^2 = \mathcal{E}^2 - f S \left(1 + \frac{l^2}{r^2} - \beta^2\right) \,, \tag{20}$$

$$\dot{\phi} = \frac{l}{S r^2} \,. \tag{21}$$

One can define the effective potential of radial motion of magnetized particles at equatorial plane as

$$\dot{r}^2 = \mathcal{E}^2 - 1 - 2V_{\text{eff}} \,, \tag{22}$$

where the effective potential has the following form

$$V_{\text{eff}} = \frac{1}{2}\left[f \, S \left(1 + \frac{l^2}{r^2} - \beta\mathcal{L}[\lambda_{\hat{\alpha}}]\right) - 1\right] \,, \tag{23}$$

where $\beta = 2\mu B_0/m$ is the magnetic coupling parameter. In the limiting case when $L = 0$ the expression (23) coincides with Schwarszchild one (see e.g., [42]), when $L = 0$ and $\beta = 0$ it describes the radial motion of test particles around Schwarszchild black holes [42].

Now we consider the circular orbits which satisfy the following conditions

$$\dot{r} = 0 \,, \qquad \frac{\partial V_{\text{eff}}}{\partial r} = 0 \,. \tag{24}$$

Solving the first equation of (24) one may find the expression for $\beta$ in the form

$$\beta(r) = \frac{1}{\mathcal{L}[\lambda_{\hat{\alpha}}]}\left(1 + \frac{l^2}{r^2} - \frac{\mathcal{E}^2}{f \, S}\right) \,. \tag{25}$$

Using the expression for effective potential (23) and the second equation of (24) one may write the following relation

$$\frac{\partial V_{\text{eff}}}{\partial r} = f \, S \mathcal{L}[\lambda_{\hat{\alpha}}]\frac{\partial \beta}{\partial r} \,. \tag{26}$$

The interaction term between magnetized particle and external magnetic field in equatorial plane takes the following form

$$D \cdot F = 2\mu B_0 f \, e^{\Psi} \,, \tag{27}$$

where $e^{\Psi} = \left[S\left(f - \Omega^2 r^2\right)\right]^{-\frac{1}{2}}$ and $\Omega$ is the angular velocity of the particle in the reference frame. Comparing Equations (27) and (18) one can easily find

$$\mathcal{L}[\lambda_{\hat{a}}] = e^{\Psi} f S .\tag{28}$$

The angular velocity can be found in the following form

$$\Omega = \frac{d\phi}{dt} = \frac{d\phi/d\tau}{dt/d\tau} = \frac{f}{r^2}\frac{l}{\mathcal{E}} .\tag{29}$$

Finally, the magnetic coupling parameter $\beta(r)$ can be expressed as

$$\beta(r) = \sqrt{\frac{1}{Sf} - \frac{l^2}{\mathcal{E}^2 S^2 r^2}}\left(1 + \frac{l^2}{r^2} - \frac{\mathcal{E}^2}{fS}\right) .\tag{30}$$

The expression (30) for $\beta$ is a radial function of various parameters $l$, $\mathcal{E}$, $L$, $N$. Below we will make numerical analysis of parameter $\beta$.

In Figure 3 we show the radial dependence of parameter $\beta$ for the different values of conformal factors $L$ and $N$. The left panel corresponds to the Schwarzschild black hole ($L = 0$) and plots are shown for the different values of angular momentum of magnetized particles for the fixed values of energy $\mathcal{E} = 0.9$. In the central and right panels of Figure 3 the radial dependence of parameter $\beta$ is shown for the different values of conformal parameters $L$ and $N$, for the fixed values of energy and angular momentum of particles. From the Figure 3 one can easily see that the value of the $\beta$ parameter has a minimum near the position $r = 3M$ for the value of $l$ greater than $\sqrt{5}$. With the increase of both parameters of conformal gravity the minimum and maximum values of the parameter $\beta$ also increase. In case of $l^2 \geq 6$ the minimum of the graphs disappears at high values of parameter $N$ due to the modification of the magnetic field structure near the horizon for the high values of the conformal parameter $N$ (see Figure 2). From the results on radial dependence of magnetic coupling function for the different values of conformal parameters $L$ and $N$ presented in the Figure 3 one may conclude that the existence of the nonvanishing conformal parameters increases the gravitational potential around the central object. Consequently, the magnetic coupling parameter is responsible for the strength of the interaction between magnetized particles and external magnetic field which amplifies the gravitational field of compact object. This causes the stable circular orbits shift to the direction of observer at the infinity. In our previous works we have shown that the critical values of the $\beta$ parameters decreases with the increase of the values of tidal brane charges and negative deformation, while the increase the positive deformation causes the decrease of the $\beta$ parameter. Existing region of stable circular orbits shifts to the observer at the infinity with the increase negative deformation parameter and brane charges [44,46].

The conditions for the stable orbits for the magnetized particles can be expressed by the following equations:

$$\beta(r, l, \mathcal{E}, L, N) = \beta ,\tag{31}$$
$$\beta'(r, l, \mathcal{E}, L, N) = 0 ,\tag{32}$$

where the prime $'$ denotes the partial derivative with respect to radial coordinate. The Equations (31) and (32) can be used to determine the minimum values of the function $\beta$ which corresponds to the minimum value of (specific) energy $\mathcal{E}$:

$$\mathcal{E}_{\min}^2 = f\frac{\mathcal{A}_1 - \sqrt{\mathcal{A}_1^2 - 12\mathcal{A}_2\mathcal{A}_3 fl^2 rS^2}}{6\mathcal{A}_3 r^2 S} ,\tag{33}$$

where

$$\mathcal{A}_1 = \left(L^2 + r^2\right)\left\{l^2\left[S^2(2r - 3M) + r\right] + Mr^2S^2\right\} - 2fNL^2r\left[l^2\left(S^2 + 4\right) + r^2S^2\right], \quad (34)$$

$$\mathcal{A}_2 = (3 - 4N)l^2 + \left[3l^2 + (1 - 4N)L^2\right]r^2 + r^4, \quad (35)$$

$$\mathcal{A}_3 = L^2(2Nfr + M) + Mr^2. \quad (36)$$

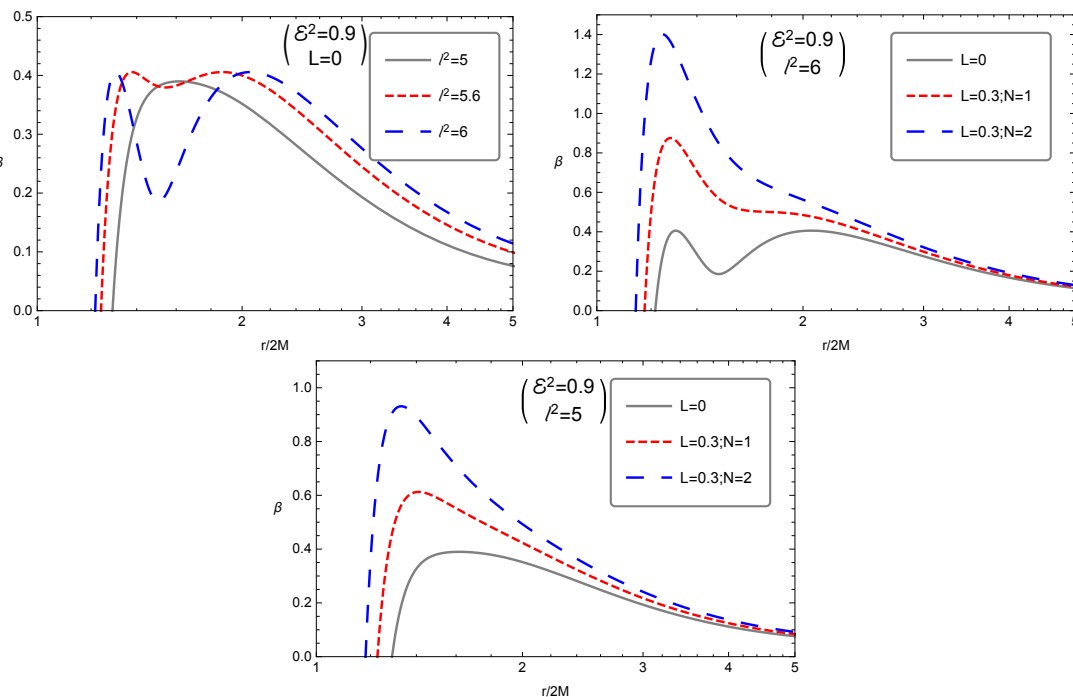

**Figure 3.** Radial dependence of magnetic coupling function for the different values of conformal parameters $L$ and $N$. The **top-left panel** corresponds to the Schwarzschild black hole ($L = 0$) and in the **top-right** and **bottom panels** correspond to the cases of effects of conformal gravity for the different values of angular momentum of magnetized particles when the energy is fixed as $\mathcal{E} = \sqrt{0.9}$.

In the limiting Schwarzschild case, we have $L = 0$ ($S = 1$) and the energy takes the following form

$$\mathcal{E}_{\min} = \frac{l(r - 2M)}{\sqrt{M}r^{3/2}},$$

which corresponds to the known result of Ref. [42].

Figure 4 illustrates the radial dependence of minimum energy for the different values of the conformal parameters $L$ and $N$. From the Figure 4 one can easily see that the value of minimum energy decreases with the increase of conformal parameters $L$ and $N$ near the event horizon.

Now we start to analyze the minimum value of parameter $\beta = \beta_{\min}(r, l, \mathcal{E}, L, N)$ which can be found using Equation (33) together with Equations (34)–(36)

$$\beta_{\min} = \frac{\sqrt{\mathcal{A}_1^2 - 12\mathcal{A}_2\mathcal{A}_3fl^2rS^2} - \mathcal{A}_1 + 6\mathcal{A}_3rS^2\left(l^2 + r^2\right)}{6\mathcal{A}_3\sqrt{f}r^3S^{5/2}}\left[1 + \frac{6\mathcal{A}_3l^2r}{\sqrt{\mathcal{A}_1^2 - 12\mathcal{A}_2\mathcal{A}_3fl^2rS^2} - \mathcal{A}_1}\right]^{1/2} \quad (37)$$

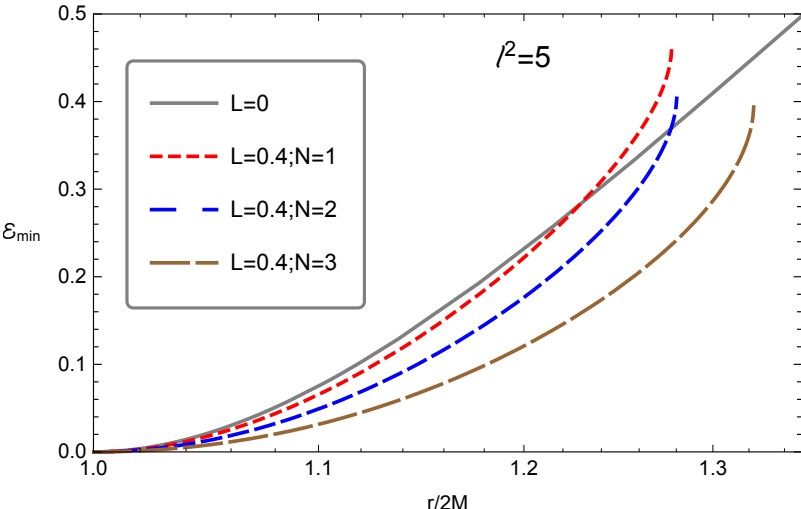

**Figure 4.** Radial dependence of the minimum energy of the particle for the different values of conformal parameters $L$ and $N$ with the specific angular momentum $l^2 = 5$. The value of minimum energy decreases with the increase of conformal parameters $L$ and $N$ near the horizon.

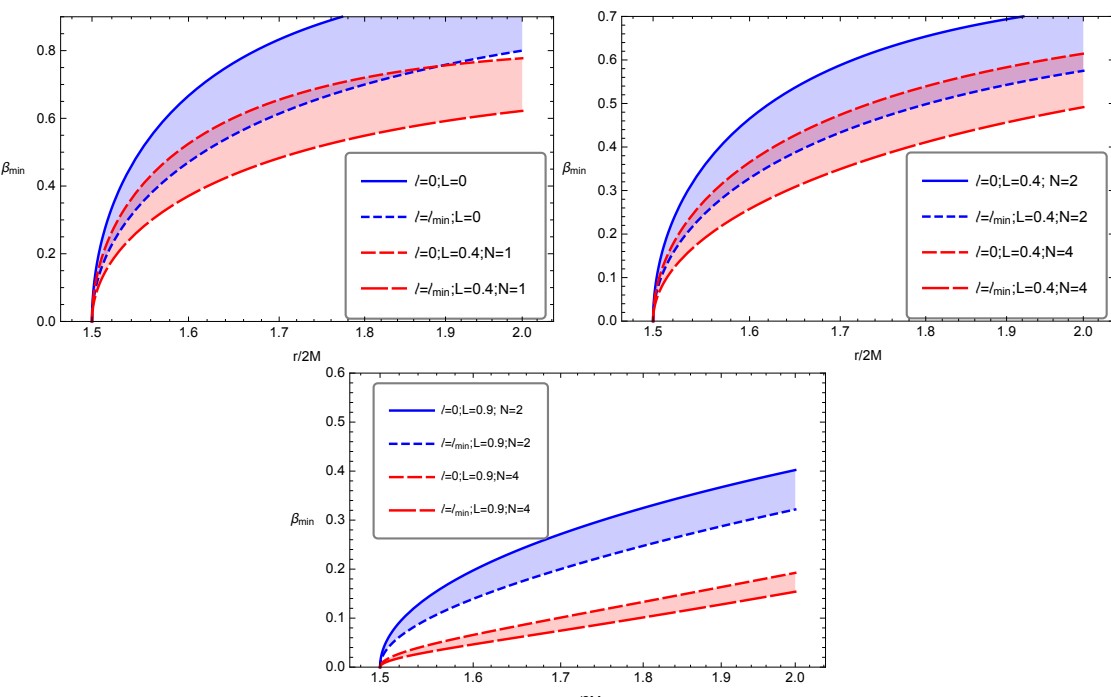

**Figure 5.** The radial dependence of the minimum value of parameter $\beta$ for the different values of the conformal parameters $L$ and $N$. The colored region represents the range for the values of $\beta$ in which stable circular orbits are allowed for magnetized particles for $r > 3M$. The values of upper and lower limits of the parameter $\beta_{min}$ decrease with the increase of conformal parameters $L$ and $N$. In **top-left panel** the blue and the light-red colored areas correspond to the case of Schwarzschild black hole and the values of the conformal parameters $L = 0.4M$ and $N = 1$, respectively. In **top-right** and **bottom panels** the blue and the light-red colored areas correspond to the values of the conformal parameter $L = 0.4M$ and $L = 0.9M$ for the fixed values of the parameter $N = 2$ and $N = 4$, respectively.

To see the effect of conformal parameters $L$ and $N$ we present the radial dependence of the $\beta_{min}$ for the different values of conformal parameters in Figure 5. In the Figure 5 the colored area corresponds to the set of $r$ and $\beta$ parameters of stable circular orbits of magnetized particles. The borders of the shaded area corresponds to the values of the parameter $\beta$ at $l = 0$ and $l = l_{min}$ which are found using

the condition $\partial \beta_{min} / \partial r = 0$. One can see from the Figure 5 that values of upper and lower limits of the parameter $\beta_{min}$ decreases with the increase of conformal parameters $L$ and $N$. Moreover, the shaded area slightly shrinks with the increase of the value of the conformal parameter $N$, while with the increase of the value of parameter $L$ the shaded area significantly reduces.

## 4. Magnetized Particles Acceleration around Regular Black Holes Immersed in an External Magnetic Field

In this section, we will study the energetic processes around black holes in conformal gravity in the presence of magnetic field. Specifically, we consider collision of two particles near the black hole in conformal gravity immersed in external magnetic field. We study the effect of conformal parameter and external asymptotically magnetic field to the center-of-mass energy of two colliding particles coming from infinity with energies $\mathcal{E}_1$ and $\mathcal{E}_2$. The center-of-mass energy of two colliding particles can be expressed as

$$\mathcal{E}_{cm}^2 = \frac{E_{cm}^2}{2mc^2} = 1 - g_{\alpha\beta} v_1^\alpha v_2^\beta \, , \tag{38}$$

here $v_1^\alpha$ and $v_2^\alpha$ are the velocities of two particles with the energy at infinity $\mathcal{E}_1$ and $\mathcal{E}_2$, respectively. Below we consider several scenarios when the particles are (i) both magnetized, (ii) magnetized and charged (iii) magnetized and neutral, (iv) both charged at the equatorial plane where $\theta = \pi/2$ and $\dot{\theta} = 0$.

### 4.1. Collision of two Magnetized Particles

In this subsection we will consider the center-of-mass energy of colliding two magnetized particles with the equal mass and magnetic dipole moment around a non-rotating black hole immersed in an external asymptotically uniform magnetic field in conformal gravity. The expression for center-of-mass energy of two magnetized particles can be derived substituting the Equations (19)–(21) into (38) in the following form

$$\mathcal{E}_{cm}^2 = 1 + \frac{\mathcal{E}_1 \mathcal{E}_2}{f\,S} - \frac{l_1 l_2}{S\,r^2} - \sqrt{\mathcal{E}_1^2 - f\,S\left(1 + \frac{l_1^2}{r^2} - \beta_1^2\right)} \cdot \sqrt{\mathcal{E}_2^2 - f\,S\left(1 + \frac{l_2^2}{r^2} - \beta_2^2\right)} \, . \tag{39}$$

Figure 6 illustrates the radial dependence of the center-of-mass energy of two colliding magnetized particles for the different values of the conformal parameters $L$ and $N$. One can see from the Figure 6 that the increase of the values of both conformal parameters $L$ and $N$ cause to decrease of the value of the center-of-mass energy. The effect of the parameter $N$ on the decreasing rate of the center-of-mass energy is much stronger than the effect of parameter $L$. Moreover, the minimum value of collision distance decreases with increasing both conformal parameters $L$ and $N$.

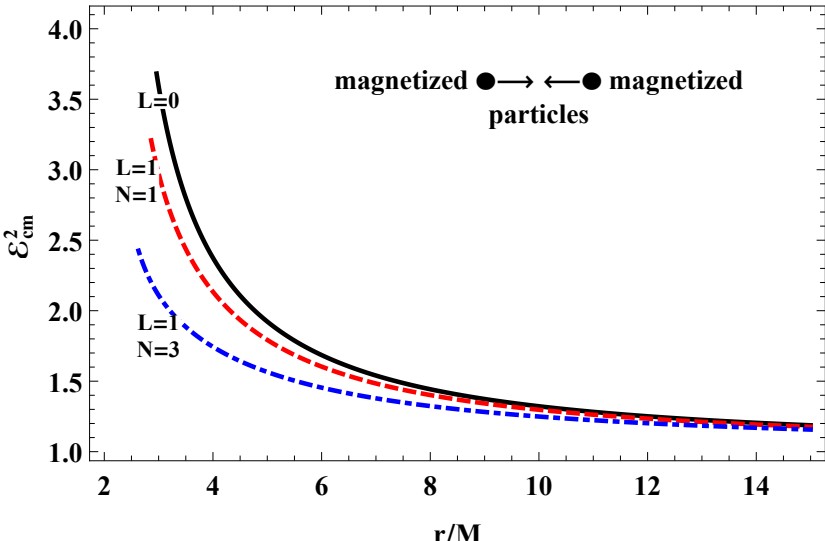

**Figure 6.** Radial dependence of center-of-mass energy of two colliding magnetized particles near the black hole in conformal gravity. The plots have been produced for the values of $l_1 = -l_2 = 2$, $\mathcal{E}_1 = \mathcal{E}_2 = 1$ and $\beta_1 = \beta_2 = 1$. Conformal parameters $L$ and $N$ cause to decrease of the value of the center-of-mass energy of the colliding particles.

*4.2. Collision of Magnetized and Charged Particles*

Now, we will study the collision of the magnetized particle with charged one. To derive the expression for center-of-mass energy first we must derive the four-velocities for charged particles. It can be found using the Lagrangian for the charged particle

$$\mathcal{L} = \frac{1}{2}m\,g_{\mu\nu}u^\mu u^\nu + e\,u^\mu A_\mu \,, \tag{40}$$

where $e$ is electrical charge of the particle.

The energy and the angular momentum of the charged particles can be found in the following form

$$p_t = \frac{\partial \mathcal{L}}{\partial \dot{t}} \implies -E = mg_{tt}u^t \,, \tag{41}$$

$$p_\phi = \frac{\partial \mathcal{L}}{\partial \dot{\phi}} \implies L = mg_{\phi\phi}u^\phi + eA_\phi \,, \tag{42}$$

and the components of four-velocity of the charged particle are

$$\dot{t} = \frac{\mathcal{E}}{Sf} \,, \tag{43}$$

$$\dot{r} = \sqrt{\mathcal{E}^2 - f\left[S + \left(\frac{l}{r} - \omega_B rS\right)^2\right]} \,, \tag{44}$$

$$\dot{\phi} = \frac{l}{Sr^2} - \omega_B \,, \tag{45}$$

where $\omega_B = eB/(2mc)$ is cyclotron frequency of the charged particle in the uniform magnetic field.

The expression for center-of-mass energy of magnetized and charged particles inserting equations of motion of charged particles (19)–(21) and (43)–(45) to the expression (38) in the following form

$$\mathcal{E}_{cm}^2 = 1 + \frac{\mathcal{E}_1\mathcal{E}_2}{fS} - \frac{l_1 l_2}{Sr^2} - \sqrt{\mathcal{E}_1^2 - f(r)\left[S + \left(\frac{l_1}{r} - \omega_B rS\right)^2\right]} \cdot \sqrt{\mathcal{E}_2^2 - fS\left(1 + \frac{l_2^2}{r^2} - \beta^2\right)} \,. \tag{46}$$

Figure 7 shows the radial dependence of center-of-mass energy of colliding (positive and negative) charged and magnetized particles with different values of the conformal parameters $L$ and $N$. One can see from the Figure 7 in the case of collision of the negative charged and magnetized particles value of the center-of-mass energy decreases with increasing the values of the conformal parameters $L$ and $N$. Moreover, increasing of the values the conformal parameters $L$ and $N$ the energy does not cause the high energy of colliding particles in the far region. The point where the center-of-mass energy is maximum comes closer to the central object in the case of the colliding magnetized and negatively charged particles; however, in the case of the collision of magnetized and positively charged particles the point is the same for all values of the conformal parameters.

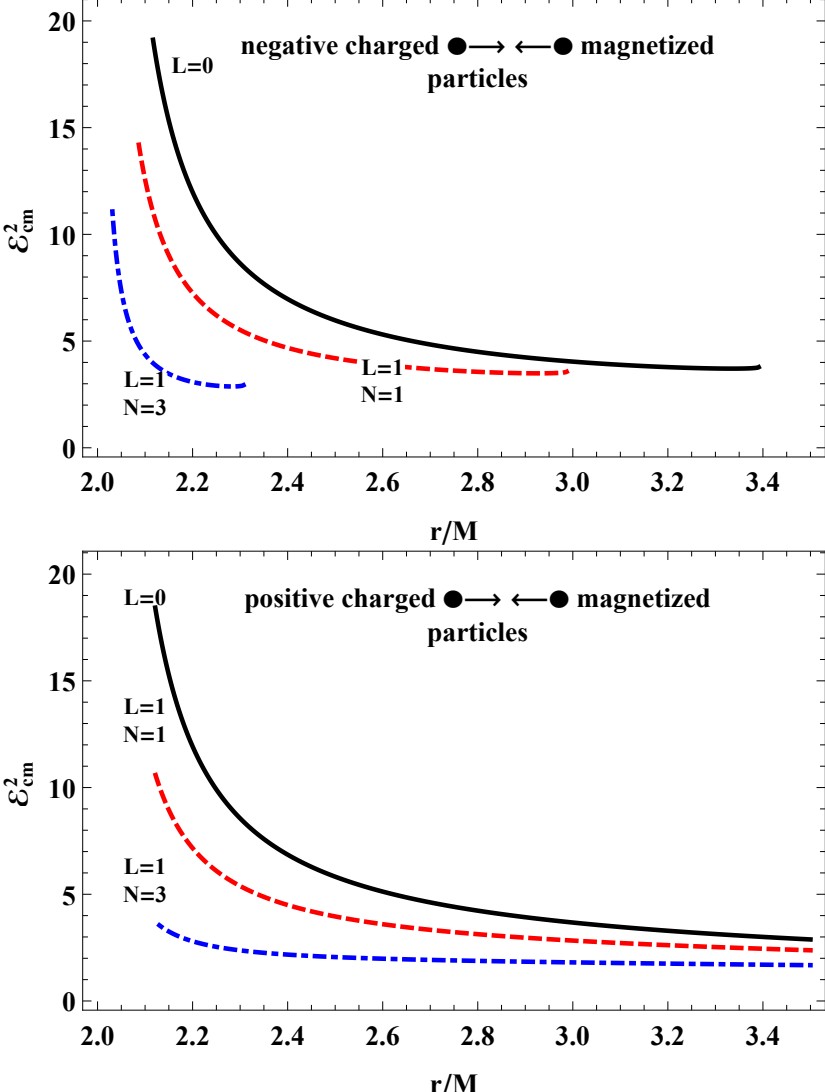

**Figure 7.** Radial dependence of center-of-mass energy of collision of charged and magnetized particles for the different values of conformal parameters $L$ and $N$. The plots have been taken for the values of $l_1 = -l_2 = 2$, $\mathcal{E}_1 = \mathcal{E}_2 = 1$ and $\beta = 1$ and $\omega = \pm 0.5$. In the case of collision of the negative charged and magnetized particles value of the center-of-mass energy decreases with increasing of the values of the parameters $L$ and $N$. The **top** and **bottom** panels show the collision of magnetized particle with negatively and positively charged particles, respectively.

### 4.3. Collision of Two Magnetized and Neutral Particles

This subsection is devoted to study of the effects of conformal gravity on center-of-mass energy of neutral and magnetized particle collision.

The expressions for equations of motion of neutral particles in conformal gravity can be written in the following form

$$\dot{t} = \frac{\mathcal{E}}{fS}, \tag{47}$$

$$\dot{r} = \sqrt{\mathcal{E}^2 - fS\left(1 + \frac{l^2}{r^2}\right)}, \tag{48}$$

$$\dot{\phi} = \frac{l}{Sr^2}. \tag{49}$$

The expression for center-of-mass energy of the collision between two magnetized particles can be derived in the standard way substituting Equations (19)–(21) and (47)–(49) to Equation (38) as following

$$\mathcal{E}_{cm}^2 = 1 + \frac{\mathcal{E}_1\mathcal{E}_2}{fS} - \frac{l_1l_2}{Sr^2} - \sqrt{\mathcal{E}_1^2 - fS\left(1 + \frac{l_1^2}{r^2} - \beta_1^2\right)} \cdot \sqrt{\mathcal{E}_2^2 - fS\left(1 + \frac{l_2^2}{r^2}\right)}. \tag{50}$$

Figure 8 shows radial profiles of the center-of-mass energy of colliding neutral and magnetized particles. One can see that the maximum value the center-of-mass energy decreases as the increase of the conformal parameters and the point where the energy extraction is maximum comes closer in the higher values of the conformal parameters.

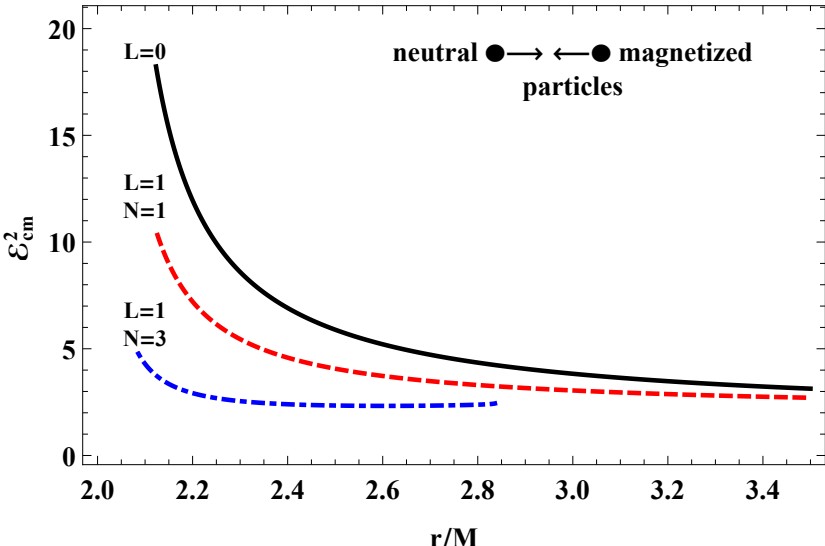

**Figure 8.** Radial dependence of center-of-mass energy of colliding neutral and magnetized particles for the different values of conformal parameters $L$ and $N$. The plots are taken for the values of $l_1 = -l_2 = 2$, $\mathcal{E}_1 = \mathcal{E}_2 = 1$ and $\beta_1 = 1$. The center-of-mass energy decreases with the increase of the conformal parameters.

### 4.4. Collision of Two Charged Particles

Finally, we will consider the collision of two charged particles. The expression for center-of-mass energy of the two charged particles around the black hole in the uniform magnetic field in conformal gravity, can be obtained substituting Equations (43)–(45) into Equation (38) as

$$\mathcal{E}_{cm}^2 = 1 + \frac{\mathcal{E}_1\mathcal{E}_2}{fS} - \frac{l_1l_2}{Sr^2} - \sqrt{\mathcal{E}_1^2 - f\left[S + \left(\frac{l_1}{r} - \omega_B r S\right)^2\right]} \cdot \sqrt{\mathcal{E}_2^2 - f(r)\left[S + \left(\frac{l_2}{r} - \omega_B r S\right)^2\right]}. \tag{51}$$

Figure 9 shows the radial dependence of the center-of-mass energy of colliding two charged (positive–positive, negative–negative and positive–negative) particles with different values of the conformal parameters $L$ and $N$, considering the initial energy of both colliding charged particles, mass and the absolute value of electric charge of the particles are the same. One can also see that the energy decreases with increasing the value of both $L$ and $N$. On the other hand, the dependence of the energy on radial coordinate looks slightly different for collisions of positive–positive, negative–negative and positive–negative charged particles. The analysis of the particles acceleration near the black hole in conformal gravity shows that the increase of the gravitational potential energy of the central object results in the change of the center-of-mass energy of the colliding particles. However, the innermost collision position does not depend on the parameters of the conformal gravity. This is related to the fact that innermost collision point coincides with the photon sphere and conformal parameters do not affect the photon sphere [12,13]. Consequently, the inner position of the collisions point remains unchanged in the conformal gravity.

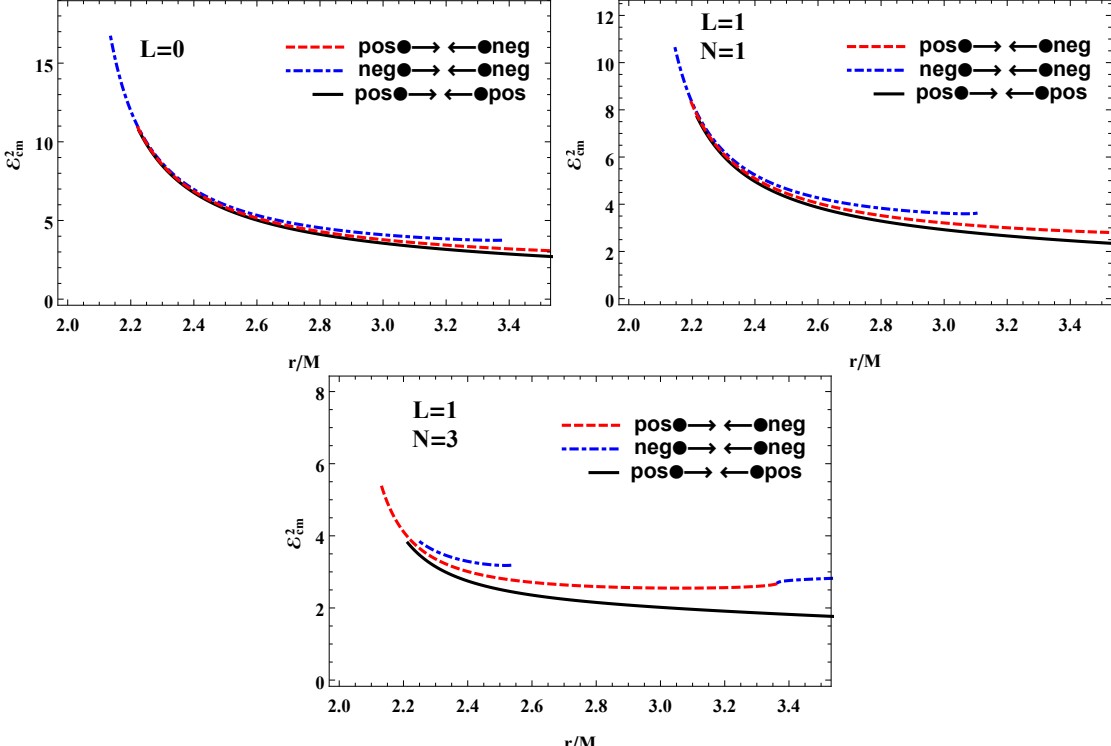

**Figure 9.** Radial dependence of center-of-mass energy of two colliding charged particles for the different values of conformal parameters $L$ and $N$. The plots are taken for the values of $l_1 = -l_2 = 2$, $\mathcal{E}_1 = \mathcal{E}_2 = 1$ and $|\omega| = 0.5$. The energy decreases with increasing the values of both conformal parameters $L$ and $N$. **Left-top panel** shows the both positive and negative charged particles around Schwarzschild black hole ($L = 0$). **Top-right** and **bottom panels** correspond to the cases when the value of the parameter $N = 1$ and $N = 3$ at the fixed values of the parameter $L = M$, respectively.

## 5. Astrophysical Applications

There is great interest in relativistic astrophysics to probe black hole physics i.e., to measure its main parameters as a spin and mass.

The measurements of mass of supermassive black holes Sgr A* at the center of Milky Way is straightforward and detected with high precision in infrared wavelength through analysis of motion of S star around Sgr A* in the Newtonian gravity framework.

The measurements of spin of the supermassive black hole is much more difficult. The most close $S_2$ is orbiting Sgr A* at the minimum distance about 1000 Schwarzschild radii where the angular velocity of dragging of inertial frames due to the decay as $1/r^3$ with compare to its value at the super

massive black hole event horizon is weaker for $10^9$ times and it is impossible to measure it. Other way to measure the spin of supermassive black holes at the center of Milky Way is through precise measurement of pulsars motion [60,61]. Pulsars are highly magnetized neutron stars and can be treated as magnetized test particles in supermassive black hole environments.

We will study here possible astrophysical scenario of pulsar motion in the in supermassive black hole environment and make rough estimation of orbital radius change due to interaction of magnetic moment of the neutron star with the magnetic field in supermassive black hole environments. As an example we will consider the typical neutron star with the magnetic field $B = 10^{12}$ G of mass $M_{NS} = 1.4 M_\odot$, where $M_\odot$ is the solar mass, moving around supermassive black hole with the total mass $M = 10^6 M_\odot$, i.e., $M_{NS} \ll M$. The external magnetic field can be taken to be equal to $10^2 G$ [62].

The estimation of the $\Delta r$ can be expressed as

$$\Delta r = 2.733 \cdot 10^3 \text{ cm} \left( \frac{B_{NS}}{10^{12} \text{ G}} \right)^2 \left( \frac{B_{BH}}{10^2 \text{ G}} \right)^2 \left( \frac{M}{10^6 M_\odot} \right) \left( 1 + \frac{L^2}{10^{12} M_\odot^2} \right)^{-\frac{19}{2} N}. \tag{52}$$

One may conclude that the size of the area where stable circular orbits of magnetized particles are allowed shrinks with the increase of both conformal parameters $L$ and $N$. In the previous works it was shown that in the presence of the brane tidal charge and negative deformation of space-time around the black hole the area expands, while in the case of the presence of the positive deformation the area shrinks [44,46] which is similar to the effect by the parameters of conformal gravity. One may study ISCO radius of magnetized particle around black hole immersed in an asymptotically uniform magnetic field using the standard way as $V_{\text{eff}} = \mathcal{E}, V_{\text{eff}} = 0, V_{\text{eff}} \geq 0$.

ISCO radius of a particle around rotating black hole can be calculated through the expression

$$r_{isco} = 3 + Z_2 - \sqrt{(3 - Z_1)(Z_1 + 2Z_2 + 3)}, \tag{53}$$

where $Z_1 = \left( \sqrt{1-a} + \sqrt[3]{a+1} \right) \sqrt[3]{1 - a^2} + 1, Z_2 = \sqrt{3a^2 + Z_1^2}$.

Figure 10 shows the comparison of ISCO radius of test particles around rotating Kerr black hole and one for magnetized particles around non-rotating black hole immersed in external magnetic field. One can see that presence of magnetic field and magnetized particle motion can mimic the motion of test particle around Kerr black hole (see Figure 11) which means that the degeneracy between spin of black holes and the effect of magnetic field in black hole environments.

Figure 11 illustrates the degeneracy between spin of black holes and magnetic interaction in black hole environments. It shows that for the cases when $\beta < 1$ the degeneracy with the reasonable value of black hole spin up to 0.6 becomes very realistic.

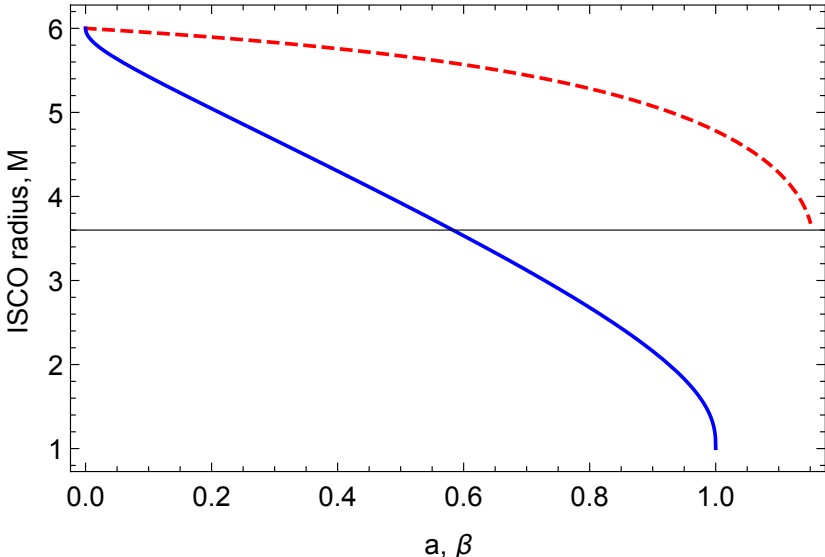

**Figure 10.** Comparisons of ISCO radius of the neutral particles around rotating Kerr black holes with one for magnetized particles around Schwarzschild black holes immersed in external magnetic field. Red dashed line corresponds to the dependence of ISCO radius of magnetized particle on magnetic parameter $\beta$ while blue solid line corresponds to the dependence of test particles on rotation parameter $a$.

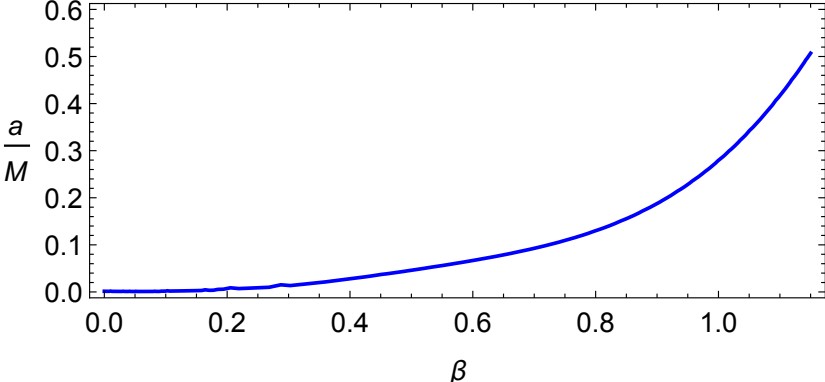

**Figure 11.** The degeneracy plot showing the dependence of rotation parameter $a$ on $\beta$. The line corresponds to the matching values of parameters for the same values of ISCO.

## 6. Conclusion

The study of magnetized particle motion around compact objects can play an important role to test gravity theories in the strong gravitational field regime. In this work we have studied the motion of magnetized particles and energetic processes in the vicinity of magnetized non-rotating black holes in conformal gravity. The obtained results can be summarized as follows

- We have analyzed the behavior of magnetic field near the event horizon of the black hole in conformal gravity. It is shown that with the increase of conformal parameters $L$ and $N$ the value of angular component of magnetic field at the stellar surface decreases. Moreover, the increase of the conformal parameters forces the asymptotically uniform magnetic field lines to be more curved and for the large values of $N$ ($N \gg 1$) the structure of asymptotically uniform magnetic field turns to be a dipolar-like near the event horizon of black holes.
- The maximum value of the effective potential corresponding to circular motion of the magnetized particle increases with the increase of conformal parameters. However, the minimum value of the particle's specific energy as well as upper and lower limits for minimum values of $\beta$ parameter decrease near $r = 3M$.

- The colored region in Figure 5 indicating the range between upper and lower limits of the $\beta_{min}$ parameter sufficiently shrinks at higher values of conformal parameter $L$, while the effect of the parameter $N$ to the shrinking is not sufficient.
- Analysis of two colliding particles near the black hole environment shows that in all cases of neutral, charged and magnetized particle collision the center-of-mass energy decreases with the increase of conformal parameters $L$ and $N$.
- It is shown in Figure 7 that the property/behavior of the center-of-mass energy differs in the case of magnetized particle colliding with positive and negative charged particles in conformal gravity. For instance, in the case of the collision of magnetized and negatively charged particles the innermost collision point comes closer at higher values of the parameters of conformal gravity. However, there is no observed dependence of the collision point on the values of the conformal parameters $L$ and $N$.
- We have applied the obtained results to the real astrophysical scenario when a pulsar treated as a magnetized particle is orbiting the supermassive black hole (SMBH) Sgr A$^*$ in the center of our galaxy in order to obtain the estimation of magnetized compact object's orbital parameter. The possible detection of pulsar in Sgr A$^*$ close environment can provide constraints on black hole parameters. Here we have shown that the interaction of magnetic field $\sim 10^2$ Gauss with magnetic moment of magnetized neutron star can in principle mimic spin of Kerr black holes up to 0.6.

**Author Contributions:** Conceptualization, K.H., A.A., J.R. and B.A.; methodology, J.R.; software, K.H., J.R.; validation, B.A., A.A. and A.A.; formal analysis, K.H., A.A.; investigation, K.H., A.A., J.R., B.A. All authors have read and agreed to the published version of the manuscript.

**Funding:** This research is supported by Grants No. VA-FA-F-2-008, No. MRB-AN-2019-29 and No. YFA-Ftech-2018-8 of the Uzbekistan Ministry for Innovative Development, and by the Abdus Salam International Centre for Theoretical Physics through Grant No. OEA-NT-01. This research is partially supported by an Erasmus+ exchange grant between SU and NUUz.

**Acknowledgments:** Authors thank Silesian University in Opava for hospitality.

**Conflicts of Interest:** The authors declare no conflict of interest.

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
