# Peer review of "Magnetized Particle Motion around Black Holes in Conformal Gravity: Can Magnetic Interaction Mimic Spin of Black Holes?"

_universe, doi:10.3390/universe6030044_

Round 1

Reviewer 1 Report

In this manuscript, authors have studied magnetized particle motion around black hole in conformal gravity immersed in asymptotically uniform magnetic field. Moreover, they also have showed the real astrophysical scenario when a pulsar treated as a magnetized particle is orbiting the supermassive black hole Srg A in the center of our galaxy in order to obtain the estimation of magnetized compact object's orbital radius. 

In my opinion, the paper is well conceived and well presented. which adds to the present knowledge of the subject. The motivation is good and english is reasonably good. Hence I recommend that this article may be published in the standard journal Universe.

Author Response

Thank you very much for spending time to review our submission. We have improved English of our submission in the revised version.

Reviewer 2 Report

The essay displays serious flaws. It deals with the motion of a charged particle around a Schwarzschild black hole immersed in an external magnetic field, but it applies the results to a real astrophysical case as Sgr A*, which in turn is a rotating black hole. Authors also refer to gravitational collapse during the elaboration of the theory, but this can apply only to stellar-mass black holes and does not apply to a supermassive black hole as Srg A*. Therefore, the proposed theory cannot be applied to the case of Sgr A*.The Equation (53) is wrong. The essay also shows many typos, missing references, and not all the adopted symbols are explained. 

Author Response

We would like to thank all Referees 1, 2 and 3 for the careful read of the manuscript, useful comments and suggestions. In the revised version of the manuscript we have taken into consideration all points arisen by both Referees 2 and 3. We hope the revised version of the paper performed according to the Referees comments and suggestions will be quite straightforward.
Below please find our detailed answers to the comments of the Referees.

Reply to the Referee 2

Comment: It deals with the motion of a charged particle around a Schwarzschild black hole immersed in an external magnetic field, but it applies the results to a real astrophysical case as Sgr$A^*$, which in turn is a rotating black hole. Authors also refer to gravitational collapse during the elaboration of the theory, but this can apply only to stellar-mass black holes and does not apply to a supermassive black hole as Srg$A^*$. Therefore, the proposed theory cannot be applied to the case of Sgr$A^*$. The Equation (53) is wrong. The essay also shows many typos, missing references, and not all the adopted symbols are explained.

Answer: Thanks to the Referee B for the useful comments and questions. In the manuscript we are not studying the gravitational collapse of the object at all. We simply represent the discussions of other authors on the reasons why the black hole cannot have intrinsic magnetic field concluding that the stellar magnetic field will decay and vanish during the gravitational collapse of the object. The theory and effects considered in the paper are not related to the collapse mechanisms.

The main message of the paper is to show that magnetic field surrounded compact object may, in principle, mimic the rotation parameter of the black hole. Considering the magnetized particles motion around non-rotating compact object embedded in external magnetic field in conformal gravity, one may mimic the rotation parameter of Kerr black hole up to $0.6$. Due to this reason we have applied our results to the real astrophysical scenario of the close environment of Sgr A*. You are absolutely right concerning correctness of the Equation (53), unfortunately it was wrong due to the typo made. Now it is corrected.

Reviewer 3 Report

In this paper, the magnetized particle motion around black hole is investigated in conformal gravity for asymptotically uniform magnetic fields . In particular, the behavior of magnetic fields near the horizon of the black hole is explored and it is shown that with the increase of conformal parameters, $L$ and $N$, the value of angular component of magnetic field at the stellar surface decreases. The discussions are interesting and the mathematical results might be helpful for the related future works. Moreover, the manuscript is written in detail. Thus, if the following points are reconsidered carefully, this paper could be worthy of being published.

1. There would exist a number of the past related works on the magnetized particle motion around black hole in the literature. By comparing with these preceding studies, the new ingredients and significant progresses of this work should be stated more explicitly and in more detail. That is, the differences between this paper and the past ones should be described in more detail and more clearly. This is the most crucial point in this review.

2. What are the astrophysical reasons why the maximum value of the effective potential corresponding to circular motion of the magnetized particle increases with the increase of the conformal parameters?

3. For all cases of neutral, charged and magnetized particles collisions the center-of-mass energy decreases with the increase of conformal parameters $L$ and $N$. Furthermore, in the case of the collision of magnetized and negatively charged particles, the inner most collision point comes closer at higher values of the parameters of conformal gravity and the collision point does not depend on the values of the conformal parameters. From these facts, what can we learn on the astrophysical consequences on conformal gravity?

4. It is shown that the interaction of magnetic filed $\sim 10^2$ Gauss with magnetic moment of magnetized neutron star can in principle mimic spin of Kerr black hole up to 0.6. From this result, is it possible to obtain any new information of the astrophyiscal natures of the supermassive black hole Srg12 A∗?

5. Finally, it is recommended that the wordings and grammar of English should be rechecked throughout the present manuscript.

Author Response

We would like to thank all Referees 1, 2 and 3 for the careful read of the manuscript, useful comments and suggestions. In the revised version of the manuscript we have taken into consideration all points arisen by both Referees 2 and 3. We hope the revised version of the paper performed according to the Referees comments and suggestions will be quite straightforward.
Below please find our detailed answers to the comments of the Referees.

Reply to the Referee 3

Comment 1: There would exist a number of the past related works on the magnetized particle motion around black hole in the literature. By comparing with these preceding studies, the new ingredients and significant progresses of this work should be stated more explicitly and in more detail. That is, the differences between this paper and the past ones should be described in more detail and more clearly. This is the most crucial point in this review.

Answer: Thanks to the Referee 1 for his useful comments regarding comparisons of the obtained results with past works related to the magnetized particle motion. Taking into account the Referee’s comment we have included the following text into the revised version of the manuscript.

In our previous works we have shown that the critical values of the $\beta$ parameters decreases with the increase of the values of tidal brane charges and negative deformation, while the increase the positive deformation causes the decrease of the $\beta$ parameter. Existing region of stable circular orbits shifts to the observer at the infinity with the increase negative deformation parameter and brane charges (see Refs.(Rahimov11,Rayimbaev16))

One may conclude that the size of the area where stable circular orbits of magnetized particles are allowed shrinks with the increase of both conformal parameters $L$ and $N$. In the previous works it was shown that in the the presence of the brane tidal charge and negative deformation of spacetime around the black hole the area expands, while in the case of the presence of the positive deformation the area shrinks (see Refs.(Rahimov11,Rayimbaev16)) which is similar to the effect by the parameters of conformal gravity.

Comment 2: What are the astrophysical reasons why the maximum value of the effective potential corresponding to circular motion of the magnetized particle increases with the increase of the conformal parameters?

Answer: Thanks to the Referee A for the useful comment regarding the physics behind the effect of circular orbits modification due to effects of the conformal parameters. Taking into account the Referee's comment we have included the following text into the revised version of the manuscript.

From the results on radial dependence of magnetic coupling function for the different values of conformal
parameters $L$ and $N$ presented in the figure 3 one may conclude that the existence of the nonvanishing conformal parameters increases the gravitational potential around the central object. Consequently, the magnetic coupling parameter is responsible for the strengthetness of the interaction between magnetized particles and external magnetic field which amplifies the gravitational field of compact object. This causes the stable circular orbits shift to the direction of observer at the infinity.

Comment 3: For all cases of neutral, charged and magnetized particles collisions the center-of-mass energy decreases with the increase of conformal parameters $L$ and $N$. Furthermore, in the case of the collision of magnetized and negatively charged particles, the inner most collision point comes closer at higher values of the parameters of conformal gravity and the collision point does not depend on the values of the conformal parameters. From these facts, what can we learn on the astrophysical consequences on conformal gravity?

Answer: Thanks to the Referee for the useful comment regarding the role of conformal prameters effect on energetic process around compact object. Taking into account the Referee's comment we have included the following explanations into the revised version of the manuscript.

The analysis of the particles acceleration near the black hole in conformal gravity shows that the increase of the gravitational potential energy of the central object results in the change of the center of mass energy of the colliding particles. However, the innermost collision position does not depend on the parameters of the conformal gravity. This is related to the fact that innermost collision point coincides with the photon sphere and conformal parameters do not affect to the photon sphere [BAMBI-ref]. Consequently, the inner position of the collisions point remains unchanged in the conformal gravity.

Comment 4: It is shown that the interaction of magnetic filed $\sim 10^2$ Gauss with magnetic moment of magnetized neutron star can in principle mimic spin of Kerr black hole up to 0.6. From this result, is it possible to obtain any new information of the astrophyiscal natures of the supermassive black hole Srg $A^*$?

Answer: Thanks to the Referee A for the comment regarding possible degeneracy between the magnetic coupling parameter and spin of the black hole.

One can conclude that in the observations of the magnetized particles motion around the super massive black holes, in the case when the black hole rotation parameter is less than 0.6, the effects of the magnetic field around the black hole and the spin parameter of the black hole on the ISCO of the magnetized particle are not distinguishable.
Thus one can not conclude measuring the ISCO radius whether its shift is due to rotation effect or the effect of the external magnetic fields.

Comment 5: Finally, it is recommended that the wordings and grammar of English should be rechecked throughout the present manuscript.

Answer: Taking into account the Referee's useful comment we have edited the grammar of the text as much as we can do it.

Round 2

Reviewer 3 Report

The authors' answers to the review report are appreciated very much.
In the revised manuscript, the points suggested in the review report have been reconsidered. Thus, this paper can be accepted for publication in Universe.